# Bacteria Living in Biofilms in Fluids: Could Chemical Antibiofilm Pretreatment of Culture Represent a Paradigm Shift in Diagnostics?

**DOI:** 10.3390/microorganisms12020259

**Published:** 2024-01-26

**Authors:** Lorenzo Drago, Andrea Fidanza, Alessio Giannetti, Alessio Ciuffoletti, Giandomenico Logroscino, Carlo Luca Romanò

**Affiliations:** 1Laboratory of Clinical Microbiology, Department of Biomedical Sciences for Health, University of Milan, 20133 Milan, Italy; 2UOC Laboratory of Clinical Medicine, MultiLab Department, IRCCS Multimedica, 20138 Milan, Italy; 3Mininvasive Orthopaedic Surgery—Department of Life, Health and Environmental Sciences, University of L’Aquila, 67100 L’Aquila, Italy; andrea.fidanza@univaq.it (A.F.); giandomenico.logroscino@univaq.it (G.L.); 4Unit of Orthopaedics and Traumatology, “SS Filippo e Nicola” Hospital, 67051 Avezzano, Italy; 5Unit of Orthopaedics and Traumatology, “G. Mazzini” Hospital, 64100 Teramo, Italy; alessio.giannetti@graduate.univaq.it (A.G.); alessio.ciuffoletti@aslteramo.it (A.C.); 6Romano Institute, Rruga Deshmoret e 4 Shkurtit, 1001 Tirana, Albania; info@romanoinstitute.com

**Keywords:** biofilm, fluids, dithiothreitol (DTT), microbiological culture examination, diagnosis

## Abstract

Biofilms are multicellular aggregates of bacteria immersed in an extracellular matrix that forms on various surfaces, including biological tissues and artificial surfaces. However, more and more reports point out the fact that even biological fluids and semifluid, such as synovial liquid, blood, urine, or mucus and feces, harbor “non-attached” biofilm aggregates of bacteria, which represent a significant phenomenon with critical clinical implications that remain to be fully investigated. In particular, biofilm aggregates in biological fluid samples have been shown to play a relevant role in bacterial count and in the overall accuracy of microbiological diagnosis. In line with these observations, the introduction in the clinical setting of fluid sample pretreatment with an antibiofilm chemical compound called dithiothreitol (DTT), which is able to dislodge microorganisms from their intercellular matrix without killing them, would effectively improve the microbiological yield and increase the sensitivity of cultural examination, compared to the current microbiological techniques. While other ongoing research continues to unveil the complexity of biofilm formation in biological fluids and its impact on infection pathogenesis and diagnosis, we here hypothesize that the routine use of a chemical antibiofilm pretreatment of fluid and semi-solid samples may lead to a paradigm shift in the microbiological approach to the diagnosis of biofilm-related infections and should be further investigated and eventually implemented in the clinical setting.

## 1. Introduction

Planktonic microorganisms are characterized by their ability to develop in culture media, and they are usually regarded as freely suspended cells. Although, as early as the seventeenth century, Antonie Philips van Leeuwenhoek documented the existence of surface-associated microbes that develop and reside in communities. The biofilm, a typical community organization of life developed by bacteria, differs greatly from that of their planktonic state. A bacterial biofilm is a colony of ordered bacteria enclosed in a self-manufactured extracellular matrix, which is made up of proteins, DNA, and polysaccharides [1]. These matrices offer a safe haven for the residing bacteria, encouraging cooperation and communication and empowering the community to react to environmental changes as a whole. Almost any surface, biotic or abiotic, can support the formation of this unusual and intricate biological system [2].

One important initial step in the creation of biofilms is the adhesion of bacteria to surfaces. Adherence can happen on artificial surfaces like medical devices as well as on biological surfaces like host tissues. Bacterial adhesins are surface features that aid the interaction [3]. The first adhesion of planktonic bacteria to a surface triggers the complex, multi-stage series of processes known as biofilm development. These bacteria transition to a surface-attached, community-based lifestyle, which is a significant change from their individual, free-swimming form. This is known as the biofilm lifestyle [4]. After the first adhesion, bacteria multiply and produce microcolonies, which eventually develop into a three-dimensional structure. Mature biofilms allow bacterial cells to thrive and survive in difficult environments by encasing them in a protective, nutrient-rich matrix. “Dispersion” is the process by which bacterial cells leave their biofilms, revert to an independent planktonic lifestyle, and eventually colonize new surfaces to create new biofilm-based communities [5]. This process allows the biofilm to spread across a surface. The surface of a medical device, for example, provides an ideal substratum for bacterial attachment and biofilm formation, leading to device-associated infections [6].

It has been observed recently that bacteria can form biofilms in fluids such as synovial fluid, urine, cerebrospinal fluid, blood, mucus, saliva, and feces [7,8,9,10,11,12,13]. Numerous environmental elements further influence the biofilm development process in biological fluids: bacterial adhesion, microcolony formation, and maturation can be impacted by variables like pH, oxygen tension, temperature, and fluid shear forces [14]. Furthermore, unique surfaces for bacterial interaction are presented in biological fluids. For example, extracellular components and host cells can act as substrates for bacterial adhesion and biofilm development and can also be influenced by the presence of other microorganisms or particulate debris in the fluid, which can either encourage or hinder the process [15].

Biofilms in biological fluids provide a serious clinical risk and a major health issue because they contribute to chronic infections that are difficult to treat and identify. Moreover, they fortify bacteria against antibiotic therapy and the immune system reaction of the host [16]. In patients with cystic fibrosis, for example, biofilm infections can lead to a variety of problems, such as bloodstream infections, lung infections, endocarditis, and urinary tract infections [17]. As a result, eliminating and managing biofilms pose serious difficulties in clinical practice [18].

The purpose of this paper is to give a general overview of the bacteria that can be found in human fluids as biofilm aggregates, with an emphasis on how the biofilm may be destroyed without killing the microorganisms by performing a chemical pretreatment of fluid samples. This may allow for a better identification of the pathogen, proper diagnosis, and treatment of the infection.

## 2. Synovial Fluid (SF)

Because periprosthetic joint infections (PJIs) require longer hospital stays, costly and sophisticated procedures, and protracted antibiotic treatments, they pose a significant burden to both patients and healthcare systems [19,20]. Arthrocentesis can show whether a joint effusion is inflammatory or not, but microbiological confirmation is needed to rule out or identify an infection. Recent reports have linked the resistance of bacterial joint infections to standard therapies to the presence of bacteria and biofilm aggregates floating in the SF [7,21,22]. With reported sensitivities as low as 45% [23], the poor efficacy of existing microbiological tests on SF may be explained by the ability of bacteria to reside in biofilm aggregates.

In this regard, since 2015, Dastgheyb et al. [21] have demonstrated that *Staphylococcus aureus* are able to form clumps in SF samples in less than 20 h, even when cefazoline is added into the samples. These clumps are organized with a biofilm structure, which was confirmed by staining with wheat germ agglutinin (WGA). WGA stains the polysaccharide coating of *Staphylococcus aureus* biofilms because of the presence of polysaccharide intercellular adhesin (PIA). Furthermore, this aggregation causes an accompanying decrease in virulence, such as with biofilm-like phenotypes.

The protein content and the viscosity (150 ± 50 MPa) of SF favors the formation of biofilms, whether anchored or floating.

On the other hand, the same behavior was not noticed in tryptic soy broth (TSB); in TSB, *Staphylococcus aureus* maintained its planktonic state, and was effectively eradicated by low doses of cefazolin.

The same year, in a Letter to the Editor, Perez et al. [22] declared similar results obtained by culturing *Staphylococcus epidermidis* not in human SF, but in bovine SF.

In a recent work by Drago et al. [7], it was shown that the ability of bacteria to live in aggregates in the SF may be a common behavior among several if not all bacteria. They collected SF from 57 subjects affected by a painful total hip or knee replacement, with suspicions of a PJI; then, the samples were divided into two aliquots, one pretreated with dithiothreitol (DTT), and one with normal saline.

DTT is a sulfhydryl compound able to reduce disulfide bonds between polysaccharides and neighbor proteins, acting as an antibiofilm agent without any toxicity to the living bacteria. The final finding of this study was that DTT pretreatment of SF samples from patients with PJIs allowed us to improve the pathogen count and cultural examination sensitivity, when compared to a control group with a normal saline solution. In addition, positive samples or higher bacterial counts were found not only when the pathogen was *Staphylococcus aureus*, but also for other *Staphylococci*, *Proteus mirabilis*, *E. coli*, and *E. faecalis.*

As occurs with bacteria living in biofilms attached to surfaces, the creation of fluctuating biofilm aggregates may also hinder the ability of antibiotics to reach and kill the microbial cells. Additionally, the phenotypic and virulence factor production in these aggregates is altered by the bacteria–protein interactions, leading to a notable antibiotic tolerance [24].

For this reason, the presence of bacteria in PJI synovial fluids may play a significant role in the emergence of a chronic illness that is challenging to identify and manage; therefore, the use of effective antibiofilm pretreatments, as was just discussed, could be effective in increasing cultural examination sensitivity and yield [7].

Figure 1 shows a self-coaggregation of bacteria biofilm embedded in synovial fluid, while Figure 2 discloses a coaggregation of bacteria (A) and the single bacteria cells after dithiothreitol (DTT) pretreatment (B).

## 3. Urinary Tract

In the context of urinary tract infections (UTIs), biofilm formation on urinary catheters or on uroepithelial cells leads to recurrent infections that are less responsive to standard antibiotic therapies. Biofilms easily develop on urinary catheters due to various factors. The characteristics and surface topology of the catheter, as well as the presence of irregularities and surface striations, facilitate the initial attachment of bacteria. Latex catheters may also have embedded diatom skeletons that act as attachment sites for bacteria.

Once a urinary catheter is inserted, a conditioning film forms on its surface. This film is derived from urine constituents and host proteins, such as fibrinogen, and supports bacterial adhesion and the formation of biofilms.

In long-term catheterization, where patients are catheterized for four weeks or more, the catheters are exposed to contaminated urine for an extended period. This increases the likelihood of bacterial colonization and rapid biofilm formation, making newly inserted catheters susceptible to quick contamination.

Uropathogenic *Escherichia coli*, the leading causative agent of UTIs, is well-known for its biofilm-forming capabilities, often leading to treatment failure and disease recurrence [2].

As one of the most prevalent bacteria-related diseases, urinary tract infections (UTIs) pose a serious threat to public health. It is estimated that these diseases cost the US economy USD 3.5 billion annually in operating expenses. UTIs can present as prostatitis, urethritis, pyelonephritis, or cystitis, among other manifestations. Uroepithelial cells and urinary catheters are ideal sites for uropathogen adhesion and colonization. The most common agents responsible for complicated UTIs are *Escherichia coli*, *Enterococcus* spp., *Klebsiella pneumoniae*, *Candida* spp., *Staphylococcus aureus*, *Proteus mirabilis*, and *Pseudomonas aeruginosa* [26]. The aforementioned bacteria are all well known for their ability to create biofilms, which frequently results in treatment failure and illness recurrence [27].

Large biofilm particles and high microbial cell densities have the potential to break from the catheter, flow into the bladder, disseminate infection, and cause bacteriuria. Furthermore, uropathogens have the ability to grow as a biofilm in the kidney and bladder, which lowers antibiotic sensitivity and increases the risk of recurring infections. Particularly in patients with prolonged catheterization, biofilms are crucial in catheter-associated UTIs, which raise morbidity and mortality [28].

In contrast to the planktonic state, bacterial populations living in biofilms exhibit more adaptive and efficient behavior, increasing their chances of survival. Additionally, the biofilm community releases planktonic cells that may potentially infiltrate nearby tissues [8].

## 4. Blood System

Equally as relevant and concerning as UTIs are blood system infections (BSIs), which represent the 12th leading cause of death in the United States, with an estimated 15–30% mortality rate.

Biofilm formation on intravenous catheters is, in fact, a significant concern in clinical settings. When an intravenous catheter is inserted, it provides a surface for microorganisms to adhere to and form a protective biofilm. The biofilm not only promotes the survival and growth of the microorganisms and protection from antimicrobial agents, but also serves as a source for potential bloodstream infections. Microorganisms have the ability to migrate from a local infection (endocarditis, meningitis, osteomyelitis, etc.) to distant areas via the bloodstream.

The presence of biofilms on intravenous catheters can lead to catheter-related bloodstream infections (CRBSIs), which are associated with increased morbidity, mortality, and healthcare costs. Preventing biofilm formation on intravenous catheters is crucial and requires strict adherence to infection control practices, such as proper hand hygiene, aseptic techniques during catheter insertion, and regular catheter care. Additionally, the development of novel antimicrobial coatings and strategies to disrupt or remove biofilms holds promise in reducing the incidence of CRBSIs and improving patient outcomes [8].

The most frequently identified pathogens in bacteremia, systemic illness, and sepsis are all well-known biofilm-producing bacteria: *S. aureus*, *E. coli*, *K. pneuomoniae*, *P. aeruginosa*, *Enterococcus faecalis*, *Staphylococcus epidermidis*, *Enterobacter cloacae*, *Streptococcus pneumoniae*, *Enterococus faecium*, and *Acinetobacter baumannii* [8,29,30].

Similar to cultural investigations of other fluids, blood culture analyses can frequently yield erroneous negative results.

Planktonic microorganisms may enter the circulation through the biofilms that bacteria frequently build on intravascular devices such central venous catheters [29]. However, as recently reported by Vestby and coworkers [31], bacteria of the biofilm rarely enter the blood stream as planktonic bacteria, and for this reason the blood culture may be negative when testing for microorganisms. Consequently, immunodiagnostic assays (ELISA) have been developed to detect serum antibodies against biofilm matrix components. For example, an ELISA has been developed to detect antibodies against staphylococcal slime polysaccharide antigens. To date, the ELISA assays developed do not have the sensitivity and specificity to alone determine biofilm-associated infections.

## 5. Cerebrospinal Fluid (CSF)

The implantation of cerebrospinal fluid (CSF) shunts is a common procedure performed annually to alleviate cranial pressure caused by hydrocephalus. CSF is redirected through subcutaneous tubing from the cerebral ventricle to the peritoneal cavity. However, the high rate of complications associated with CSF shunts is evident, as the ratio of shunt revisions to primary shunt placements can reach 3:1 in many healthcare institutions. Infections contribute significantly to shunt failure, accounting for 5–30% of cases. These infections can manifest locally as ventriculitis or peritonitis, or, more systemically, as shunt nephritis or septicemia. They are associated with an increased risk of seizures, decreased cognitive function, and a twofold increase in long-term mortality. Most infections are caused by representatives of the skin flora; *S. epidermidis* accounts for about 50% of these infections, followed by *S. aureus*, accounting for about 25%. Shunt colonization occurs commonly during surgery and leads to symptomatic infection within 1 month [9]. The diagnosis and treatment of device-related infections pose significant challenges due to the formation of biofilms by the causative bacteria. Biofilms are communities of microorganisms that adhere to surfaces and are embedded within an extracellular matrix. Traditional diagnostic cultures, such as aspirates and swabs, often yield false-negative results because the microorganisms primarily exist in the biofilm state, with very few cells in the planktonic (free-floating) state. This makes it difficult to detect and identify the specific bacteria causing the infection.

Eradicating biofilms is also challenging due to the antimicrobial tolerance of the bacteria within the biofilm. These bacteria have undergone complex adaptations to survive in high-cell density environments and endure nutritional starvation. As a result, they exhibit reduced susceptibility to antimicrobial agents, making treatment less effective.

Additionally, the extracellular matrix of the biofilm inhibits the bactericidal activity of inflammatory host cells. This means that even if the immune system responds to the infection, it may struggle to effectively eliminate the bacteria within the biofilm.

Addressing device-related infections requires innovative approaches that target biofilms specifically. This may involve the development of novel antimicrobial agents or strategies that disrupt the biofilm structure and enhance the efficacy of antimicrobial treatments. Improved diagnostic techniques that can accurately detect biofilm-associated infections are also needed to guide appropriate treatment decisions [32,33].

## 6. Saliva

Through the use of a simple microscope, Van Leeuwenhoek made the first observation of bacteria on the dental plaque that covers tooth surfaces, an example of biofilm in the oral cavity [2].

Later, with the advance of technology, by employing a combination of super-resolution confocal imaging and scanning/transmission electron microscopy, it was revealed that microorganisms in human saliva were seldom observed in a solitary, free-living state; instead, they formed clusters, some of which were associated with desquamated oral epithelial cells. The microbial aggregates present in saliva exhibited a size range from 3 to 10 mm, up to 50 mm in diameter. Two distinct subpopulations of microbial structures have been identified in saliva, categorized by size: a minor fraction (<3 μm) and a major fraction (>3 μm). Microscopic examination disclosed that the fraction under 3 μm primarily comprised single cells. In contrast, the large fraction (>3 μm) exhibited considerable heterogeneity, encompassing aggregates composed of diverse bacterial cells and desquamated epithelial cells with adhered microorganisms. A cell count analysis in each fraction demonstrated that only 3% of microorganisms existed as free-living cells, while a striking 97% were present in the aggregated form. Specific taxa can be found in the aggregates such as Porphyromonas, Fusobacterium, and Haemophilus, whereas other species can be found as both free-living single cells and in aggregated communities, including Prevotella, Neisseria, Streptococcus, and Veillonella [34].

Normally, saliva is sterile until it enters the oral cavity through the salivary duct and is rapidly contaminated by microorganisms that form biofilms. The majority of infections in the throat, nose, and ears have been linked to bacterial development in the form of biofilms [10].

Bacterial adhesion and biofilm formation can occur on implanted biomaterials and other inert surfaces with weak human defenses, such as salivary calculi. In these environments, the immune system and pharmacological treatments have minimal impact on the bacteria residing in biofilms. Chronic infections of the adenoid tissue, chronic otitis media, and mastoiditis have all been linked to biofilm formation. Planktonic bacteria have the ability to break free from mature biofilms and produce acute infections like recurring and chronic otitis media. Additionally, because saliva in the mouth is contaminated with the oral microbiome [10,34], which presently forms a substantial number of biofilms, microbial diagnostics in the oral cavity are usually challenging. In this regard, it has been recently shown that polymicrobial aggregates living in human saliva effectively build the biofilm and “shape polymicrobial communities at various spatial and taxonomic scales” [34].

## 7. Tracheal Aspirate

About 25% of all nosocomial infections are pneumonia, which also appears to be the primary cause of infection in Intensive Care Units (ICUs). Nosocomial pneumonia also increases treatment expenses, the length of hospital stays, and mortality. Actually, the risk of infection increases six to twenty times when tracheal intubation is performed on a patient receiving mechanical ventilation. Recently, bacterial diagnosis using specialized specimen brushes, bronchoalveolar lavage, and endotracheal aspirates has been standardized; yet, because it relies on identifying bacteria growing in tracheal secretions, it lacks specificity. It is critical to keep in mind that biofilm is crucial to both the diagnosis and management of ventilator-associated pneumonia (VAP). The lower respiratory tract can be directly colonized by bacteria from dental plaque and the oropharynx, which naturally form biofilms, thanks to the endotracheal tube. Subsequently, bacteria within the biofilm have the ability to infect the lungs in multiple ways: by aspirating aerosolized planktonic pathogens detached from the biofilm into deeper airways, or by separating biofilm parts that eventually reach the lungs [34,35].

In recent years, studies on VAP caused by microorganisms have focused on the oropharynx and the mouth as the pathogens’ sources. *Acinetobacter baumannii* is the most frequently isolated bacterial species in tracheal secretion cultures of patients with VAP; other pathogens frequently associated with VAP are *Staphylococcus aureus*, *Pseudomonas aeruginosa*, and *Enterobacteriaceae.*

However, other pathogens such as *Citrobacter koseri*, *Proteus mirabilis*, *Pseudomonas aeruginosa*, and *Pseudomonas fluorescence* have also been detected in both oral and tracheal samples of intubated or tracheotomy patients in ICUs, indicating their important role in the pathogenesis of VAP.

The “non-attached biofilm aggregate” concept has been the focus of a detailed review just published by Kragh and coworkers from the Costerton Biofilm Center at the University of Copenhagen, in which they describe, among other things, the common occurrence of free-floating communities of bacteria in lung sputum and mucus [36].

## 8. Feces

The presence of biofilm-producing bacteria in human stools, as well as their antibiotic resistance, has been recently reported in a three-center study [12]. This study’s findings are in line with the previous observations of Bollinger and co-workers, who, more than 15 years ago, described the existence of biofilm communities in the gut [37] and with that, demonstrated the existence of bacteria aggregates in diarrhea [38]. The clinical impact of biofilms on chronic inflammatory bowel disease is still a matter of debate and thus represents a promising research field [39,40,41,42]

The ability of bacteria to live in aggregates in intestinal fluids and in fecal material may explain the limits of current microbiological culture analyses in identifying pathogens and the full spectrum of microorganisms in a given sample. This has an impact on the diagnosis of acute gastroenteritis [42,43,44].

## 9. Drains

Closed-suction drains are utilized in several surgical specialties to prevent hematoma and the collection of fluid. However, a growing amount of research from several surgical specialties indicates that drains are not always beneficial; rather, they may be needless or even detrimental, increasing the risk of infections and other wounds complications [45].

With the growing utilization of prosthetic devices in various surgical fields, the formation of biofilms has emerged as a progressively significant challenge. Drains, classified as short-term implantable devices, are susceptible to biofilm development, leading to infection risks comparable to those associated with long-term implantable devices.

The unexpectedly swift occurrence of biofilm formation on in vivo drains constitutes a noteworthy observation. Correspondingly, in vitro investigations appear to corroborate these findings. It was observed in one study, in fact, that *Staphylococcus epidermidis* exhibited a doubling time of 17 to 38 min in full growth medium during in vitro examinations of staphylococcal biofilm formation on diverse implant surfaces. Similar happenings were observed for *Staphylococcus aureus* biofilm formation [45].

This in vivo study on drains underscored substantial biofilm formation commencing as early as 2 h post drain insertion. Although no biofilm was evident in the two clinical controls, a few isolated cocci within fibrin clumps adhering to the drain surface were observed. This result implies that drains become contaminated at an early stage. 

Moreover, numerous studies have demonstrated that the use of drains increases the risk of wound infection. When employed with a prosthesis, the heightened risks associated with periprosthetic infection and its related complications argue against their use [46]. Taking into account the perfect culture medium that clotted blood is, along with the addition of a foreign entity in the form of the drain, biofilm formation may clearly occur very quickly [13]. As for prevention, even for this reason, drains should be used as little as possible—if at all feasible, for no more than 24 h—while being well supervised.

## 10. Is It Possible to Detach Bacteria That Live in Floating Aggregates in Fluids?

Nowadays, it is commonly accepted that bacteria can live in two different ways: as planktonic organisms or as surface-attached biofilms. Medical microbiologists have begun to emphasize, meanwhile, that suspended bacterial aggregates or “non-attached biofilm aggregates” constitute a significant portion of the bacterial communities found in chronic infection sites in recent years [45]. Although this third lifestyle is starting to attract a lot of attention in clinical studies, diagnostic tools in daily practice to break the aggregates’ structure and identify and fight the bacterium are still lacking.

The main purpose of routine diagnostic procedures, including culture-based methods, is to find planktonic (free-floating) germs. Because planktonic and biofilm-associated bacteria differ from one another inherently, these techniques frequently miss biofilms or misjudge their number [7]. Bacteria in biofilms, for example, can become metabolically dormant and therefore “unculturable” using traditional methods. Moreover, bacteria may be shielded from culture media by the protective biofilm matrix, producing false-negative results [47]. The ability of biofilm-associated bacteria to withstand antibiotic treatment and host immune responses is one of their distinguishing characteristics. This is because the bacteria live in a unique physiological state within the protective biofilm matrix [48].

Effective ways to treat infections linked with biofilms are urgently needed, given the therapeutic significance of biofilm formation in biological fluids. These tactics generally seek to stop the formation of biofilms, break up existing biofilms, and increase the sensitivity of microorganisms associated with biofilms to antibiotic treatments [49,50].

Different techniques are available for disrupting the bacterial biofilm permitting the culture of the germs, and these can be physical methods, such as the sonication of chemicals such as DTT. Sonication has shown to notably increase the sensitivity of microorganism identification [51] and many authors have documented the superiority of sonication in comparison with tissue culture methods, with a lower sensitivity for the latter (ranging from 61 to 76%) when compared to sonicated implants (77–95%). Previous studies have shown that DTT is a reliable alternative to sonication for microbiological diagnoses of orthopedic infections and may be even more sensitive than sonication towards *S. epidermidis*, which is often involved in peri-implant infections [7,52]. Indeed, DTT has become an important diagnostic tool for illnesses linked to biofilms [53,54,55]; in fact, disulfide bonds have been known to be broken by reducing agents like sulfhydryl compounds such as DTT, yet they are a crucial part of many biological structures, such as the extracellular matrix that keeps biofilms together. The intricate combination of proteins, exopolysaccharides, and extracellular DNA that makes up the extracellular matrix in biofilms protects bacterial cells from external agents and enhances their ability to withstand antibiotics. It is the disulfide bonds that stabilize many of these matrix components. Because DTT has reducing properties, it can dissolve these connections, spreading the biofilm and effectively breaking down the matrix.

The processing of sputum samples from individuals suffering from respiratory diseases has made extensive use of DTT. It functions as a mucolytic agent, helping to extract and identify embedded microorganisms by liquefying the thick sputum. The use of DTT can greatly improve the diagnostic yield in chronic respiratory infections such as cystic fibrosis and chronic obstructive pulmonary disease (COPD), since biofilms are a prevalent feature in these disorders [56].

Biological fluids, such as joint aspirates, are frequently obtained in orthopedic infections in order to conduct diagnostic tests. However, it is possible that the bacteria in these fluids are organized into biofilms, which could cause false-negative results when using standard culturing techniques. These samples can be processed with DTT, which will break down biofilms and increase the production of bacterial cultures [7]. Additionally, DTT has the ability to liquefy thick materials, making it easier to extract embedded microorganisms.

Considering the effects that biofilms have on human health and the role that is being discovered more and more of biofilm non-attached aggregates in fluids to promote and maintain chronic infections, further research is needed to confirm the ability of chemical antibiofilm debonding compounds, like DTT, to improve cultural examinations of fluids and semi-solid samples. Disrupting the biofilm aggregates, in fact, frees the microorganisms in a planktonic state, leading to an increased number of planktonic culturable cells and hence of colony-forming units (Figure 3A–C).

## 11. Conclusive Remarks

The widespread prevalence and significant morbidity and mortality associated with biofilm-associated infections underscore their impact on public health. Biofilms contribute to a wide range of persistent and recurrent infections, from urinary tract infections and periodontitis to chronic wounds and implant-associated infections. Moreover, they pose a substantial economic burden due to increased healthcare costs, loss of productivity, and reduced quality of life [57].

Preventing, controlling, and especially diagnosing biofilm-associated infections is a public health priority. This requires concerted efforts at various levels, including improved infection control practices in healthcare settings, the development of biofilm-resistant materials for medical devices, effective antibiotic stewardship, applying the right diagnostic approach, and public education about biofilms and their role in infections [58].

In parallel, there is an urgent need to invest in biofilm research. As discussed previously, biofilm research is a rapidly evolving field with a multitude of unexplored avenues. Harnessing the power of modern technologies and interdisciplinary collaboration can lead to significant breakthroughs in our understanding of biofilms and the development of effective antibiofilm strategies [59].

In conclusion, bacterial biofilms in biological fluids and in medical devices represent a formidable challenge, but also an exciting opportunity for scientific discovery and innovation. While we have made significant strides in our understanding of biofilms, much remains to be learned. By unravelling the right strategies for reliable diagnoses of biofilm-related infections, it will be possible to develop innovative solutions to identify and tackle infections, ultimately improving healthcare outcomes and enhancing public health. This indeed cannot be pursued without a dedicated diagnostics approach, which always considers the presence of endowed bacteria in a biofilm. Rapid and reliable diagnoses allow for more targeted and appropriated clinical or surgical approaches.

## Figures and Tables

**Figure 1 microorganisms-12-00259-f001:**
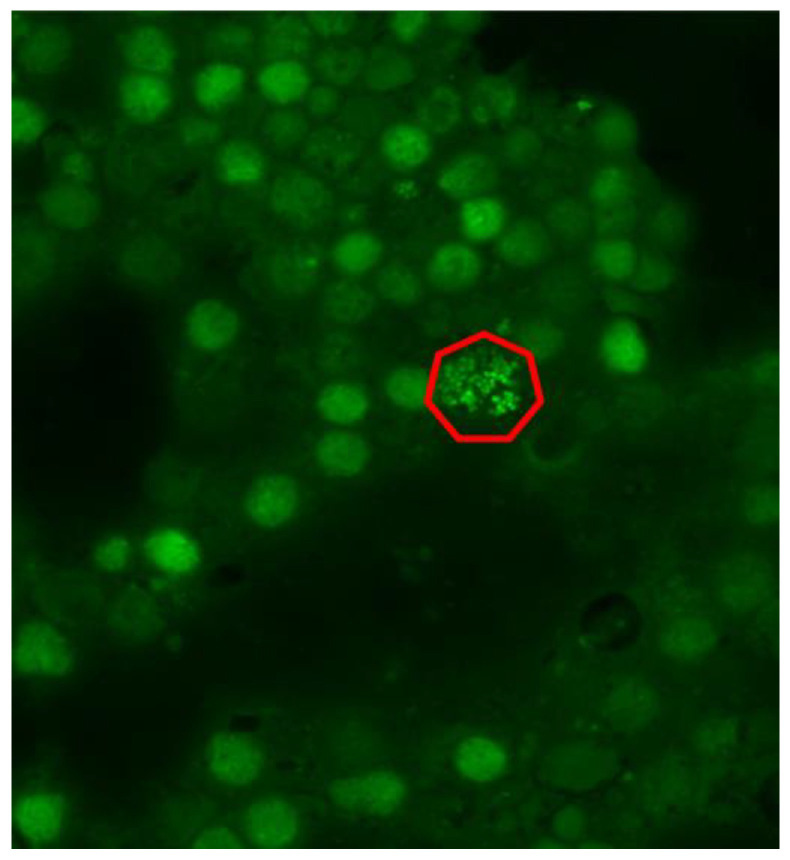
Bacteria organized in biofilm in synovial fluid. The co-aggregation of numerous bacteria (green dots) can be seen, as outlined by the red line (adapted from Bidossi et al. [25]).

**Figure 2 microorganisms-12-00259-f002:**
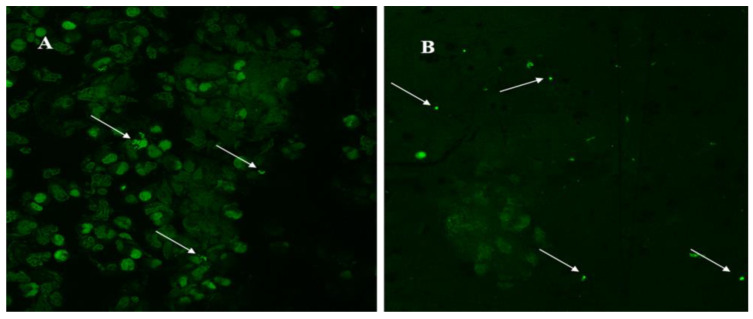
(**A**) The arrows in the figure show coaggregation of bacteria; (**B**) dithiothreitol (DTT) pretreatment breaks the biofilm structure and frees the bacteria in the planktonic state. The arrows indicate free bacteria cells after DTT antibiobilm usage.

**Figure 3 microorganisms-12-00259-f003:**
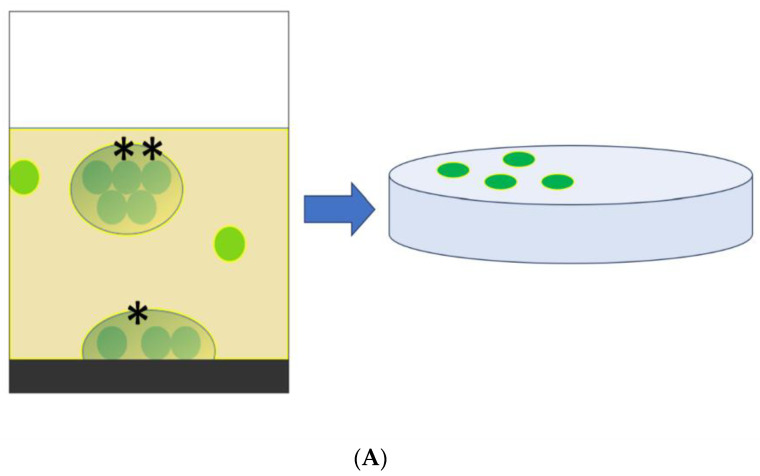
Schematic representation of the potential ability of a chemical antibiofilm pretreatment with dithiothreitol to increase the bacterial count and cultural examination sensitivity of a fluid sample: (**A**) Two planktonic bacteria (green disks) in a biological fluid sample together with biofilm-embedded bacteria adhered to a surface (*), and a non-attached biofilm aggregate (**). Cultural examination of this sample will provide two to four colony-forming units (green disks in the agar plate on the right), as each biofilm aggregate, if culturable, will be counted as only one colony-forming unit. (**B**) Dithiothreitol (red squares) added to the biological fluid breaks biofilm aggregates and frees the microorganisms in a planktonic state; (**C**) this phenomenon leads to a change in the number of planktonic culturable cells and hence of colony-forming units, which, in this example, will now be up to 10 instead of 4, as each individual cell will be able to generate a distinct colony.

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
