# Peer review of "Bacteria Living in Biofilms in Fluids: Could Chemical Antibiofilm Pretreatment of Culture Represent a Paradigm Shift in Diagnostics?"

_microorganisms, 2024, doi:10.3390/microorganisms12020259_

Round 1

Reviewer 1 Report

Comments and Suggestions for Authors

The manuscript titled »Bacteria living in biofilms in fluids: could chemical antibiofilm pre-treatment for culture represent a paradigm shift?« places its primary emphasis on investigating bacteria residing in biofilms within fluid environments, a domain that has been notably underexplored in current scientific literature. The review comprehensively aims to fill this gap in our understanding of biofilm-associated bacteria in fluidic contexts.  The overall organization of the manuscript is commendable, with both the scientific content and presentation demonstrating strength. Therefore, I recommend accepting the manuscript for publication in its current form.

Author Response

Thank you very much for your kind contribution and the appreciation of our work.

Reviewer 2 Report

Comments and Suggestions for Authors

The article “Bacteria living in biofilms in fluids: could chemical antibiofilm pretreatment for culture represent a paradigm shift?” provides a review of the literature regarding the behavior of bacteria during the formation of biofilm in biological fluids. The authors tried to describe various mechanisms of biofilm formation by bacteria. The theme of the article is very interesting and relevant. But, a few comments should be providing.

 1.  According to the title of the article, the focus is on the mechanisms of biofilm formation by bacteria from the planktonic state in biological fluids. However, processes of biofilm formation on the hard surfaces, such as urinary catheters and intravenous catheters, have also been described.

2.      Also, according to the title of the article and the purpose formulated in lines 84-88, the article should propose specific treatment methods that prevent the formation of biofilm or destroy it. However, only one method has been proposed connected with using Dithiothreitol (DTT).

3.      What is the mechanism of DTT action? Is it different for gram-positive and gram-negative bacteria?

4.      In lines 152-153 the authors wrote “Latex 152 catheters may also have embedded diatom skeletons…”. What does it mean? Diatom skeletons contained in diatomite. Diatomite is a sedimentary rock of biogenic origin. How could diatomite get into the urinary catheter?

5.      The Figure 3 is presented too schematically, a more detailed description is needed in the text of the article.

Author Response

  1. According to the title of the article, the focus is on the mechanisms of biofilm formation by bacteria from the planktonic state in biological fluids. However, processes of biofilm formation on the hard surfaces, such as urinary catheters and intravenous catheters, have also been described.

Exact. Thank you for your observation. We wanted to describe in detail the formation of biofilms even on solid surfaces, because it is from these premises that the intuition to rediscover bacteria aggregated in biofilms even in fluids begins. According to our observation bacteria could form biofilm aggregation on solid surfaces, forming then biofilm in fluids passing through the planktonic state and vice versa.

  1. Also, according to the title of the article and the purpose formulated in lines 84-88, the article should propose specific treatment methods that prevent the formation of biofilm or destroy it. However, only one method has been proposed connected with using Dithiothreitol (DTT).

Thank you for your contribution. What we wanted to convey in the title was suggesting methods for treating biological fluids in the diagnosis of infections. Early and more accurate diagnosis, minimizing false negatives even in chronic infections sustained by low-virulence bacteria, could permit to initiate therapeutic treatment as early and correctly as possible. Anyway, we modified the title if it was no clear.

About the methods to dissolve bacterial biofilm and successfully cultivate bacteria without causing damage, there are various approaches, including both physical (like sonication) and chemical methods (like DTT). We have chosen to investigate the treatment with DTT because it is a technique that we have successfully experimented in other studies and feel confident in its application [1,2,3]. We appreciate the thought-provoking suggestion; additional investigative methods could indeed be employed and compared, as has already been done in the study of biofilms on implants.

In conclusion, we appreciate the advice, and for comprehensiveness, we have added a mention with references for other sonication technique in line 392-398.

  • Drago L, RomanoÌ€ CL, Mattina R, et al. Does dithiothreitol improve bac- terial detection from infected prostheses? A pilot study. Clin Orthop Relat Res 2012;470:2915-2925. https://doi.org/10.1007/s11999-012-2415-3
  • Giannetti, A., Romano, J., Fidanza, A., Di Mauro, M., Brunetti, M., Fascione, F., & Calvisi, V. (2022). The diagnostic potential of MicroDTTect compared to conventional culture of tissue samples in orthopedic infections.Lo Scalpello-Journal36, 111-115.
  • Drago L, Signori V, De Vecchi E, et al. Use of dithiothreitol to improve the diagnosis of prosthec joint infections. J Orthop Res 2013;31:1694-1699. https://doi.org/10.1002/jor.22423

  1. What is the mechanism of DTT action? Is it different for gram-positive and gram-negative bacteria?

DTT, a sulfhydryl compound (empirical formula C4H10O2S2, MW 154.2) which is routinely        used in clinical microbiology for liquefying specimens from the respiratory tract, for the              treatment of explanted orthopedic joint prostheses, in order to assess its sensitivity and    specificity in detecting bacteria from biofilm infections, comparing results to those from sonication and periprosthetic tissue cultures. Sulfhydryl compounds are well known for their    ability to reduce disulphide bounds between polysaccharides and neighboring proteins and             to interfere with biofilm formation. Therefore, DTT can alter the extracellular matrix of biofilm   and free bacteria from it, permitting their cultivation with traditional methods [1] Both gram +        and gram – bacteria have been isolated from biofilm so we feel comfortable to assay that     there is no difference. Anyway the mechanism of actionof DTT is explained in the text in         line 392-414, we add a more detailed description than to your observation.

  • Murga R, Miller JM, Donlan RM. Biofilm formation by gram-negative bacteria on central venous catheter connectors: effect of conditioning films in a laboratory model. J Clin Microbiol. 2001 Jun;39(6):2294-7. doi: 10.1128/JCM.39.6.2294-2297.2001. PMID: 11376074; PMCID: PMC88128.

  1. In lines 152-153 the authors wrote “Latex 152 catheters may also have embedded diatom skeletons…”. What does it mean? Diatom skeletons contained in diatomite. Diatomite is a sedimentary rock of biogenic origin. How could diatomite get into the urinary catheter?

Thanks for the attention. It might actually seem strange, but it's not a typo. In fact, diatomaceous earth is used in the manufacture of catheters to prevent the catheter material from sticking to the metal forms. The source of these silica skeletons is common in latex catheters.

1-Stickler DJ, Morgan SD. Observations on the development of the crystalline bacterial biofilms that encrust and block Foley catheters. J Hosp Infect. 2008 Aug;69(4):350-60. doi: 10.1016/j.jhin.2008.04.031

2-Pelling H, Nzakizwanayo J, Milo S, Denham EL, MacFarlane WM, Bock LJ, Sutton JM, Jones BV. Bacterial biofilm formation on indwelling urethral catheters. Lett Appl Microbiol. 2019 Apr;68(4):277-293. doi: 10.1111/lam.13144

                  The Figure 3 is presented too schematically, a more detailed description is needed in the text of the article.

Thank you for your opinion, we added a more detailed explication of the figure 3 in the text.

Reviewer 3 Report

Comments and Suggestions for Authors

In this review, authors give a general overview of the biofilm aggregates in human fluids.

The authors present with figure 1 and figure 2 for biofilm in fluid. But the description about the note in each figure is unclear. How to identify coaggregation of bacteria and single bacteria should be explained in detail

The topic of this review is focusing on bacteria living in biofilms in fluids. Why there is an section about feces?

In your figure 3 legend, the word spelling of “colturable” should be double check.

Rare mechanism about how or influence on biofilms formation in biological fluids is involved in this review. If conveniences, please add this section, which may help reader to acquire more information.

Why pretreatment with DDT could inhibit the biofilm formation in fluid?

Comments on the Quality of English Language

 Minor editing of English language required

Author Response

The authors present with figure 1 and figure 2 for biofilm in fluid. But the description about the note in each figure is unclear. How to identify coaggregation of bacteria and single bacteria should be explained in detail

Thank you very much for your contribution, we modified the description of the figure 1, 2 trying to be clearer.

The topic of this review is focusing on bacteria living in biofilms in fluids. Why there is an section about feces?

Thank you for your observation. We considered feces a biological fluid, in the section we discussed about the ability of bacteria to live in aggregates in intestinal fluids and therefore in faecal material. This has an important impact on the diagnosis of acute gastroenteritis [1-2].

  1. Motta JP, Wallace JL, Buret AG, Deraison C, Vergnolle N. Gastrointestinal biofilms in health and disease. Nat Rev Gastroenterol Hepatol. 2021 May;18(5):314-334.
  2. Fraij O, Castro N, de Leon Castro LA, Brandt LJ. Stool cultures show a lack of impact in the management of acute gastroenteritis for hospitalized patients in the Bronx, New York. Gut Pathog. 2020 Jun 22;12:30.

In your figure 3 legend, the word spelling of “colturable” should be double check.

Thank you very much, we corrected the typo error.

Rare mechanism about how or influence on biofilms formation in biological fluids is involved in this review. If conveniences, please add this section, which may help reader to acquire more information.

Thank you for your opinion, we largely discussed in the introduction section the formation of biofilm on bioimplants surfaces in the lines 44-66. The formation of biofilm aggregates even in fluids is a recent intuition and the literature about it is still not rich. We described what is known in the lines 67-75.

“It has been observed recently that bacteria can form biofilms in fluids such as synovial fluid, urine, cerebrospinal fluid, blood, mucus, saliva and feces. Numerous environmental elements further influence the biofilm development process in biological fluids: bacterial adhesion, microcolony formation and maturation can be impacted by variables like pH, oxygen tension, temperature, fluid shear forces. Furthermore, unique surfaces for bacterial interaction are presented by biological fluids. For example, extracellular components and host cells can act as substrates for bacterial adhesion and biofilm development and can also be influenced by the presence of other microorganisms or particulate debris in the fluid, which can either encourage or hinder the process.”

Why pretreatment with DDT could inhibit the biofilm formation in fluid?

Thank you for your question, as far as we know, there is, in fact, no reference that dithiothreitol, when used at the concentration normally employed in the clinical setting for diagnostic purposes (0.1% or 1 g/L), has any impact on bacteria viability.

In fact, the only reference concerning an in vitro bacterial inhibition of DTT for E. coli is reported at very high concentrations of this compound, which are several times more than the concentrations used in the clinical setting [1].

1- Drago L, Romanò CL. Commentary: Dithiothreitol (DTT), When Used as Biofilm Detaching Method to Diagnose Implant-Associated Infections, Does Not Affect Microorganisms' Viability, According to the Current Literature. Front Microbiol. 2022 Feb 24;12:814945. doi: 10.3389/fmicb.2021.814945. PMID: 35345543; PMCID: PMC8957080.

Otherwise if the doubt emerges from the title, we modified it to make it clearer to the readers